# Flood Occurrence and Impact Models for Socioeconomic Applications over Canada and the United States

Manuel Grenier[1,2], Mathieu Boudreault[1], David A. Carozza[1], Jérémie Boudreault[2,3], and Sébastien Raymond[2,3]

[1]Department of Mathematics, Université du Québec à Montréal, Montréal, QC, Canada
[2]Climatic Hazards and Advanced Risk Modelling, The Co-operators General Insurance Company, Québec, QC, Canada
[3]Centre Eau Terre Environnement, Institut national de la recherche scientifique, Québec, QC, Canada

**Correspondence:** Mathieu Boudreault (boudreault.mathieu@uqam.ca)

**Abstract.** Large-scale socioeconomic studies of the impacts of floods are difficult and costly for countries such as Canada and the United States due to the large number of rivers and size of watersheds. Such studies are however very important to analyze spatial patterns and temporal trends to inform large-scale flood risk management decisions and policies. In this paper, we present different flood occurrence and impact models based upon statistical and machine learning methods over 31,000 watersheds spread across Canada and the US. The models can be quickly calibrated and thereby easily run predictions over thousands of scenarios in a matter of minutes. As applications of the models, we present the geographical distribution of the modelled average annual number of people displaced due to flooding in Canada and the US, as well as various scenario analyses. We find for example that an increase of 10% in average precipitation yields an increase of population displaced of 18% in Canada and 14% in the US. The model can therefore be used by a broad range of end-users ranging from climate scientists to economists who seek to translate climate and socioeconomic scenarios into flood probabilities and impacts measured in terms of population displaced.

## 1 Introduction

Extreme weather events such as floods consistently rank in the top three risks identified by thousands of global leaders surveyed by the World Economic Forum (Granados Franco et al., 2022). With losses increasing in large parts of the world due to rapid urbanization, aging infrastructure and climate change (see e.g., UNDRR (2022); Mazzoleni et al. (2021); Andreadis et al. (2022); Rentschler et al. (2022) for a global perspective, and Golnaraghi et al. (2020); Iglesias et al. (2021) for a Canadian and American perspective), there is mounting pressure on governments and the insurance industry to reduce the protection gap and the vulnerability of populations. Meeting these goals however rely on the ability of national and local governments as well as the financial services industry to map flood risk or acquire similar information at very large scales. Flood modelling over a country such as the United States or Canada is very challenging given the number of rivers and the size of watersheds. There is therefore a very high computational and financial cost to large-scale flood modelling leaving many stakeholders with outdated information who assume afterwards the high cost of their poor flood risk management decisions. Significant endeavors by the First Street Foundation in the United States (First Street Foundation, 2020) and the Task Force on Flood

Insurance and Relocation in Canada (Government of Canada, 2022) have provided an up-to-date view of how both countries are still vulnerable to flooding.

Fluvial or riverine flood modelling is typically broken down into top-down and bottom-up approaches. Bottom-up flood models rely on rainfall-runoff and/or hydrological models forced from statistical distributions of discharge to feed high-resolution hydraulic models (de Bruijn et al., 2014; Sampson et al., 2015; Falter et al., 2016). When used in combination with vulnerability and exposure information (e.g., damage curves, location and characteristics of properties at risk), bottom-up flood models are very useful for local governments with applications ranging from land use and emergency planning to assessing the economic viability of protection works such as dikes. But the amount of high-resolution information required and the high computational costs of numerical hydraulic models make bottom-up approaches difficult and costly for large-scale socioeconomic studies (Ward et al., 2015). On the other hand, top-down flood models rely on low-resolution hydrodynamic models forced from weather or climate models to simulate runoff or other flood-related variables of interest (Yamazaki et al., 2011; Winsemius et al., 2013). We also find applications of top-down models for future projections in Jongman et al. (2014); Dottori et al. (2018); Ward et al. (2020). Still computationally intensive, such approach is more commonly used for large-scale applications but often lack the precision and focus on the impact of flooding that bottom-up models provide that is often necessary for socioeconomic applications.

The ability to quickly run multiple what-if scenarios, integrate outputs from large ensemble weather and climate models or stochastic precipitation simulators to analyze spatial patterns and temporal trends at large-scale is very important to end-users such as portfolio managers, economists, financial analysts and actuaries to inform risk management decisions and policy. However, for both top-down and bottom-up approaches, the number of scenarios available is very limited while lacking the flexibility needed by various end-users. It is also important to emphasize that many large-scale socioeconomic studies rarely need output at the resolution provided by both top-down or bottom-up approaches and there is still a large amount of uncertainty to flood modelling (Bates, 2023), especially for rare events (beyond a return period of 50-100 years).

In this paper, we introduce statistical and machine learning models for annual (fluvial) flood occurrence and socioeconomic impact (when there is a flood), measured by population displaced, for more than 31,000 basins over all Canada and the continental United States. Behavior of the models is driven by a set of atmospheric, climatological, and socioeconomic variables as well as basin characteristics and land use. Fitting of the models and predictions from the latter are both quick (within minutes) and easily replicable on a modern desktop computer. We say the model targets socioeconomic studies because (1) it is calibrated on flood events that are significant from a socioeconomic standpoint (rather than being fitted on e.g., discharge or runoff), and (2) it directly focuses on population displaced, which is a proxy for impacts of flooding and a key driver of economic losses.

We also introduce spatial dependence in the models by assuming that downstream flood occurrence and impact may be dependent upon flood dynamics immediately upstream. Such dependence aims to mimic spatial clusters of flooding due to the overflow of water within a river and its tributaries. This is an important feature for typical insurance and reinsurance portfolio management applications that focus on the tails of the aggregate loss distribution, and therefore, on the sum of occurrence and people displaced over multiple basins. The importance of spatial dependence for large-scale flood risk assessments has been discussed by Metin et al. (2020) and modelled by Lamb et al. (2010); Wyncoll and Gouldby (2015); Quinn et al. (2019) using

copulas to correlate discharge over gauged stations (within bottom-up models). Our spatial dependence modelling approach differs widely from the latter contributions by making this dependence explicit on a set of common covariates, therefore greatly simplifying the estimation of correlations.

Our model builds on and extends Carozza and Boudreault (2021) by providing a much higher resolution view of flood risk: 31,000 basins over Canada and US versus 700. This is a necessary step for many end-users to better capture local climate dynamics and a challenging step as well for the model to remain computationally manageable for calibration, prediction and simulation purposes. Moving to higher resolution also requires to handle the flow of water upstream and downstream and as such, we introduce first order spatial autoregression in the statistical and machine learning models.

The paper is structured as follows. Section 2 presents the various datasets used to model flood occurrence and impact in terms of population displaced. Section 3 defines what is flood occurrence, presents the statistical and machine learning methods in addition to the covariates used to fit various occurrence models. Their performance is then assessed with test and validation sets and we analyze the key determinants of flood occurrence. Section 4 is structured similarly to Section 3 but for flood impact. It defines flood impact in terms of population displaced, summarizes the statistical and machine learning methods, covariates and various models, looks into the performance of the models and the key determinants of flood impact. By combining predictions of flood occurrence and impact from the previous sections, we then illustrate in Section 5 various applications of the model. Namely, an analysis of the geographical patterns of the modelled average annual population displaced due to flooding across Canada and the continental US (Section 5.1), as well as a few what-if scenarios (Section 5.2) to determine how population displaced is impacted by perturbations on precipitation. We find for example that a 10% increase in average precipitation, which is consistent with a 2-degree warming scenario, yields a larger increase in population displaced of 18% in Canada and 14% in the US. Finally, Section 6 concludes the paper with a summary of the results and a longer discussion about potential applications.

## 2 Data

Riverine (or fluvial) floods are complex phenomena that depend on an important number of variables to appropriately represent their behavior. To model flood occurrence and impact, we first gather a large set of atmospheric, hydrological, geological and socioeconomic variables. This section describes the key datasets used throughout the paper and basic operations in preparation for the statistical and machine learning methods.

### 2.1 Observed Floods

Our flood model is fitted to historical flood events that are significant from a socioeconomic point of view, meaning they typically trigger a state of emergency by a local authority or generate large (re)insurance claims. With that in mind, we choose the Global Active Archive of Large Flood Events from the Dartmouth Flood Observatory, known as the DFO database from here on (Brakenridge, 2022). This database contains significant floods from 1985 to the present. Observations in the DFO database are derived from news, governmental, instrumental, and remote sensing sources. It has been used recently by numerous authors

(Stein et al., 2020; Zhang et al., 2022; Schumann et al., 2022; Liu et al., 2022; Guido et al., 2023), notably for a wide range of applications such as flood classification and detection.

For each flood event in the DFO database, there exists a polygon that describes a raw flood extent such as a contour of areas affected (not to be confused with precise inundation extents acquired from satellite imagery for example), as well as the count of population displaced. We focus on events that took place in Canada and in the United States and the resulting dataset has 504 flood events (polygons, dates of the event and corresponding population displaced) in the 1985-2021 period. The blue shaded polygon of Figure 2 illustrates a typical observation in the DFO dataset.

While high-resolution inundation extents are increasingly becoming available (see the Global Flood Database (GFD) by Tellman et al. (2021)), as noted by Vestby et al. (2024) only a subset of the DFO events has precise extents "since adverse meteorological conditions and complex topography sometimes prevent remote sensing". Underreporting of flood events can be a concern when analyzing the frequency of events, and whether patterns and trends are statistically significant. The DFO polygons, although more rough, provide a good indicator of where and when flooding occurred, especially given our choice of geographical unit which is described next.

## 2.2 Watershed Structure

The main geographical unit used throughout this paper is the watershed. That is, flood occurrence and impact, as well as the covariates used to explain the latter are defined over watersheds. Hence, we choose the HydroBASINS database from HydroSHEDS (Lehner and Grill, 2013; Lehner et al., 2008). The HydroBASINS database is based on the Pfafstetter coding system and offers twelve different resolutions: 1 being the largest basins (size of a continent) while 12 is the smallest possible basin. For this research, we work with basins at Pfafstetter level 8 (PL8) that provide 31,410 watersheds to work with over Canada and the continental United States. This level of resolution is small enough to capture regional behaviors while being large enough so calculations are completed in a reasonable amount of time (matter of minutes up to an hour). Moreover, for each watershed the HydroBASINS dataset includes information about the basin's surface area and connection to the next downstream watershed. Figure 1 shows most of the 31,410 watersheds at Pfafstetter level 8 over Canada and the United States. The figure is truncated above the 60th parallel but the model also resolves the Canadian territories and Alaska. Moreover, we see the boundaries of PL8 watersheds in Figure 2 highlighted in red.

## 2.3 Basin Characteristics

HydroBASINS is tied to a much larger and popular database known as HydroATLAS (Linke et al., 2019). HydroATLAS offers more than 280 different variables already aggregated by watershed which can be used to e.g., better understand flood dynamics at large scales (Lindersson et al., 2021; Bernhofen et al., 2022; Chevuturi et al., 2023). This dataset also allows us to extract the same variables from upstream watersheds which is useful to create a spatial autoregressive structure in a model. After a careful examination of HydroBASINS, we selected five variables (some of which include upstream metrics) for our occurrence and impact models. These variables have been chosen based on their ability to explain water infiltration, surface runoff and

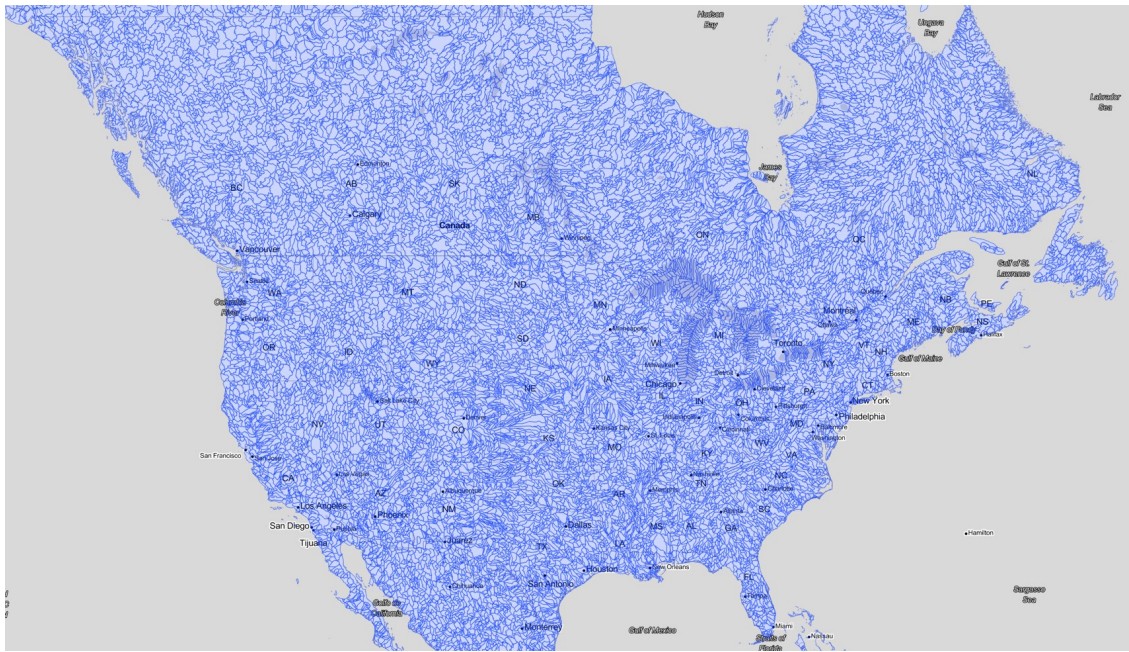

**Figure 1.** Map of Pfafstetter level 8 watersheds over Canada and the United States. Credits: Map produced using Leaflet with Stadia.StamenTonerLite layer.

the speed at which a watershed drains itself downstream. Note that variables in HydroATLAS are static, i.e. fixed at any given time.

First, the degree of regulation (`dor_pc_pva`) gives us an understanding on how structures or defences (such as dams) have an impact on the natural water flood rates (Lehner et al., 2011). It is a ratio between the volume of water contained by the watershed with and without those structures. If the ratio is above 100, then the watershed can support higher water flow rates than what it would naturally do. Secondly, stream gradient (`sgr_dk_sav`) represents the ratio between the elevation range and length of the river. This metric is constructed with the EarthEnv-DEM90 digital elevation model that has a resolution of 3 arc seconds (Robinson et al., 2014). Finally, we select three variables (along with their upstream values) to better describe soil composition (Hengl et al., 2014); that is, the percentage of sand (`snd_pc_`), silt (`slt_pc_`), and clay (`cly_pc_`) of the topsoil layer (0-5cm).

### 2.4 Atmospheric Variables and Climate Classification

Atmospheric variables such as temperature and precipitation are key elements of our proposed flood model, notably to introduce time-varying dynamics and spatial dependence. We carefully choose datasets that have a sufficiently high resolution but also have wide recognition in the climate science literature. For temperature, we choose the Daymet dataset (Thornton et al., 2021, 2020, 1997). Daymet provides the daily minimum and maximum temperatures at 1 km x 1 km resolution from 1980 and

onward across North America, including Canadian territories (Yukon, Northwest Territories and Nunavut). Data from 1985 to 2021 was extracted. Precipitation data comes from the Multi-Source Weighted-Ensemble Precipitation (MSWEP) database (Beck et al., 2019b). MSWEP provides precipitation at 3-hourly and daily frequencies from 1979 to the present at a resolution of 0.1° per 0.1°. Although it has a coarser resolution than Daymet, it has the highest accuracy in densely gauged and ungauged regions versus other known precipitation datasets such as CHIRPS and PERSIANN-CDR (Beck et al., 2019a, 2017). One shortfall of MSWEP though is its limited coverage to 60°N. To include Canadian territories in the proposed flood model, we work with Daymet's precipitation above 60°N. Precipitation is therefore aggregated per PL8 using MSWEP below 60°N and Daymet above 60°N.

For use in the statistical and machine learning methods, we build various metrics derived from temperature and precipitation. The objective is to understand how minimum and maximum temperatures and cumulative precipitation over days, weeks and months may influence riverine flooding. The first step is to find the maximum daily precipitation for each PL8 watershed and hydrological year (defined in Section 3.1). From that day, we compute the following three metrics: average daily precipitation, average daily maximum temperature and average daily minimum temperature, all three computed over the last 7 days, the previous 8-30 days, the previous 31-60 days and the previous 61-120 days. This is in line with Davenport et al. (2021) who found that 5-day and monthly state-level precipitation correlate with losses in the US. We also add longer time horizons to account for snow accumulation and melt processes that are significant over Northeastern US and most Canadian provinces. In summary, there are three atmospheric variables each measured over four non-overlapping time periods, for a total of twelve metrics.

To better capture possible interactions between temperature and precipitation, we also use the Köppen-Geiger (KG) climate classification map at a resolution of 30 arc seconds over the period 1980-2016 (Beck et al., 2018). The higher resolution is important because there are some PL8 watershed areas smaller than 1 km$^2$. The KG climate classification proposes 30 types of climates based upon various metrics of temperature and precipitation and their interactions. For each PL8 watershed, we assign the KG climate with the most significant coverage.

## 2.5 Land Use

Land use also significantly influences riverine floods (Rogger et al., 2017; UNDRR, 2022). Considering that land use is dynamic, it is imperative to find a dataset with a temporal dimension. Thus, we choose the Copernicus Global Land Cover database (Buchhorn et al., 2020), versions 2 and 3. It includes an annual raster of the land cover from 1993 to 2020 globally at a resolution of 100 m x 100 m. Since we have flood occurrences prior to 1993, we fix land use in the period 1985-1992 to that of 1993, and to that of 2020 for the year 2021. Overall, the database has 23 land use classes, which we reduce to eight generic classes (in alphabetical order):

- `Bare_ext` : Empty from vegetation, water or urbanization;

- `Crop_ext` : Agriculture, excluding forestry;

- `Forest_ext` : Large share of trees;

- `Perm_snow_ext` : Covered by snow most of the year or all year long;

- `Plain_ext` : Grass and shrub mainly;

- `Urban_ext` : Urban perimeter (residential, commercial, industrial, etc.);

- `Water_ext` : Water course, lake, etc.;

- `Wetland_ext` : Wetland.

Then for each PL8 watershed, we compute the percentage of a specific land use per year.

## 2.6   Socioeconomic Variables

Given that a flood event is included in the DFO database only when it significantly impacts a population and/or causes economic losses, it is therefore important to consider socioeconomic variables such as population and wealth in the proposed flood model. The temporal dimension of socioeconomic variables is also very important and we use data from the Socioeconomic Data and Application Center (SEDAC).

We use the Global Gridded Geographically Based Economic Data (G-Econ). It provides the gross domestic product (GDP) for the years 1990, 1995, 2000 and 2005 (Nordhaus and Chen, 2016; Nordhaus, 2006). The GDP is shown in US Dollars allowing for comparisons between countries by purchasing power parity. The resolution of each map is 100 km x 100 km. We use exponential interpolation and extrapolation per grid cell for years prior to 1990 and in between each 5-year period. We also use the Gridded Population of the World version 4 (GPWv4). It has a resolution of 30 arc seconds and has five maps covering years between 2000 to 2020. Linear interpolation and extrapolation per grid cell is applied for years prior to 2000 and in between each 5-year period. For both variables, interpolation and extrapolation follow Carozza and Boudreault (2021).

For both population and wealth datasets, values are summed per PL8 watershed. We then use GDP per capita and population density in the statistical and machine learning models.

## 3   Occurrence Model

In this section, we define flood occurrence, describe the statistical and machine learning methods with their covariates, analyze the performance of the various occurrence models and summarize the key determinants of flood occurrence from our fitted models.

## 3.1   Definition

We say we have (fluvial) flood occurrence in a given PL8 watershed when a DFO flood polygon intersects its boundaries. In practice, the intersection may be too small to be relevant, and hence, there is flood occurrence only when at least 5% of the watershed's area is covered by the DFO flood polygon. We also fitted occurrence models using a threshold of 10% instead

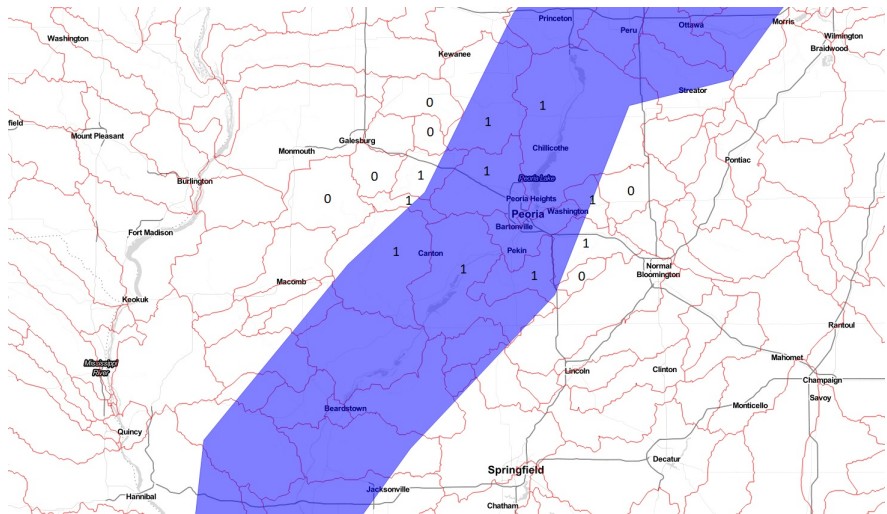

**Figure 2.** Spatial intersection of PL8 watersheds (red boundaries) with a DFO flood polygon (blue shaded area). This figure illustrates how flood occurrence is assigned in the case of a flood that took place in 1985 over Illinois, US. Ones and zeroes indicate whether an intersection was counted as a flood occurrence or not. Only a subset of the intersections are highlighted in the figure to illustrate the approach. Credits: Map produced using Leaflet with Stadia.StamenTonerLite layer.

and it had very limited impact on the results. Figure 2 shows how occurrence is computed in each basin intersected by the corresponding DFO flood polygon.

We assume that there is at most one flood occurrence per hydrological year. We choose an hydrological year rather than a calendar year to keep the seasonal precipitation cycle intact and avoid complications with respect to snow accumulation and snow melt between November and May. We follow the definition of the US Geological Survey for a hydrological year (water year, USGS (2016)) in North America; that is, a year starting on October 1st of one year and ending on September 30th of the following year.

Applying this definition of flood occurrence over Canada and the United States, we get nearly 1.2 million observations, that is 37 years of historical occurrences (ones and zeroes) for each of the 31,410 watersheds. The 504 events in the DFO thus translate into 59,651 flood occurrences or about 5.1% of observations.

### 3.2 Methodology

We defined flood occurrence as the intersection of a DFO flood polygon with a PL8 watershed (Section 3.1). Therefore, flood occurrence in this context is a classification problem with two classes: flood or no flood. Because the model is calibrated with the DFO, a flood occurrence therefore means that environmental and socioeconomic conditions are both favorable to impactful flooding.

There are two families of models one can use for classification problems: statistical and machine learning methods. The statistical learning methods we use in this paper are the Generalized Linear Models (GLM) and the Generalized Additive

Models (GAM) whereas the machine learning methods we use are tree-based techniques such as random forests (RF) and gradient boosting models (GBM). For more details about these methods, one should read Chapters 4, 7 and 8 in James et al. (2021).

**Statistical Learning Methods**

Generalized Linear Models (GLM) extend (multiple) linear regression models for responses that are not in $\mathbb{R}$. For example, the Poisson regression is a GLM model for counts (values in $\{0, 1, 2, 3, ...\}$), whereas a logistic regression is a GLM model for binary responses (either 0 or 1). Flood occurrence being a binary response, the logistic regression is therefore defined as

$$g\left(\mathbb{E}[Y]\right) = \beta_0 + \beta_1 X_1 + \beta_2 X_2 + \ldots \tag{1}$$

where $g(x) = \ln\left(\frac{x}{1-x}\right)$ is the logit function ($x \in (0, 1)$), $Y$ is a random variable taking values in $\{0, 1\}$, $X_1, X_2, ...$ are the
225 covariates (or predictors) and $\beta_0, \beta_1, \beta_2, ...$ are the coefficients of the model. Therefore the logistic regression is modelling a transformation $g$ of the probability of occurrence $\mathbb{E}[Y]$ as a linear and additive function of the predictors.

The GAM (Hastie and Tibshirani, 1990) is an extension of the GLM relaxing the assumption of linearity between the response variable and the predictors while keeping additivity of each predictor's contribution. GAM is defined as

$$g\left(\mathbb{E}[Y]\right) = \alpha + f_1\left(X_1\right) + f_2\left(X_2\right) + \ldots \tag{2}$$

where $\alpha$ is the intercept and $f_1()$, $f_2()$, ... are non-linear transformations of the covariates. Typical functions for $f$ include polynomials, logarithm, splines, etc. One can deduce that when $f$ is $\beta X$, then the GAM is a GLM.

**Machine Learning Methods**

Machine learning methods typically refer to tree-based methods as well as neural networks, among others. But since interpretability of the models is very important to our analyses, we did not use neural networks. In this paper, we therefore consider
random forests (RF) and gradient boosting models (GBM) which are both based upon decision trees (see Section 8.1 in James et al. (2021)).

RF models (Ho, 1998; Breiman, 2001) combine decision trees generated by randomly sampling observations and their features (covariates). Random forests are known to reduce overfitting and are considered a better alternative than plain decision trees. Although the relationship between the response variable and the covariates is often difficult to assess, measures such as
variable importance and sensitivity analyses allow us to better interpret the predictions from the model.

Gradient Boosting Models (GBM) are also based on decision trees but use what are known as weak learners (very small decision trees with 1 or 2 rules) to improve predictions (Friedman, 1999; Mason et al., 1999; Boehmke and Greenwell, 2019). Each weak learner tries to improve the prediction by a small amount defined by the learning rate $\lambda$. Decision trees are then sequentially added to the model to decrease the error until it reaches an optimal level of fitting. Detailed implementation of
245 both methods is provided in Appendix A.

Methods are often compared on the basis of their flexibility and interpretability. Flexibility refers to the capability of a model to fit complex non-linear relationships and interactions whereas interpretability refers to the ability of the modeller to understand and explain how a model behaves (see Section 2.1.3 in James et al. (2021)). Even if non-linearity and interactions can be included in the GLM and GAM by defining any $X$ as a function of two auxiliary covariates, it remains an arbitrary choice made by the modeller. Tree-based methods can therefore form complex interactions on their own at the price of losing some interpretability. Another advantage of GLM and GAM models is their ability to extrapolate beyond the values they have been trained on. This is useful when a model is first fitted with covariates measured under the current climate and used to make predictions with projected covariates under future climates. In such a situation, tree-based methods will make predictions based upon the closest match in the training data.

**Covariates and Models**

With the datasets described in Section 2, we can build different sets of covariates for use along with the statistical and machine learning methods. Covariates are either static (fixed in time) or dynamic (time-varying). What we refer to as the Baseline set of covariates is the following 43 covariates:

– Basin characteristics: 8 static covariates, including degree of regulation, stream gradient, clay, silt and sand fractions in soil (see Section 2.3). The upstream value is also included for the latter three.

– Atmospheric variables: 24 dynamic covariates following the 12 temperature and precipitation metrics described in Section 2.4, including their upstream values.

– Climate classification (Section 2.4): 1 static covariate with 30 factors describing the most significant climate classes;

– Land use (Section 2.5): 8 static covariates providing the percentage of the watershed covered by each type of land use;

– Socioeconomic variables (Section 2.6): 2 dynamic covariates including GDP per capita and population density;

Moreover, the Baseline set of covariates includes a dummy (indicator) variable that informs if a flood occurred in at least one of the neighboring upstream watersheds (following the definition of Section 3.1). Overall, the Baseline model includes numerous upstream variables to analyze the effect of spatial autoregression.

In addition to the Baseline model, we explore alternative formulations by including and excluding groupings of covariates. As such, models have also been fitted without upstream variables (0US suffix), that is the Baseline minus: 3 upstream basin characteristics, 12 atmospheric variables and the upstream flood dummy. The latter two sets of covariates were then used in conjunction with the GLM, RF and GBM. Moreover, our GAM models were created by taking the $\log$ of precipitation, population density and GDP per capita.

It is also important to consider interactions between covariates. Indeed, interactions allow us to determine what happens to flood occurrence (and impact, see Section 4) when two variables go in the same direction or in opposite directions. For example, when there is precipitation and temperature is below $0°$C, then snow accumulation clearly changes flood dynamics. We use

three atmospheric variables throughout the paper: maximum and minimum temperatures and precipitation. As described in Section 2.4, for each atmospheric variable, we computed averages over four non-overlapping time intervals: 1-7 days, 8-30 days, 31-60 days, and 61-120 days prior to flood occurrence. The interactions are built as follows. For each time interval, we looked at the potential interactions between the three atmospheric variables (3 possible pairs and one possible triplet), for a total of 16 possible interactions. To these 16, we add those from the upstream watershed (same combinations but from upstream) which gives us another 16 possible interactions. Therefore, the model with interactions (INTER suffix) has 32 interactions in total. Note that we only added interactions to the GLM and GAM models since by construction, the RF and GBM build such interactions empirically.

There are overall 10 models, listed as:

– GLM: 3 models, including Baseline (GLM), Baseline with interactions (GLM INTER) and Baseline without upstream variables (GLM 0US);

– GAM: 3 models, including Baseline with logged precipitation and socioeconomic variables (GAM), Baseline with interactions (GAM INTER), and the former without upstream variables (GAM 0US);

– RF: 2 models, including Baseline (RF) and Baseline without upstream variables (RF 0US),

– GBM: 2 models, including Baseline (GBM) and Baseline without upstream variables (GBM 0US).

### 3.3 Performance

Models were first fitted on a training set determined as a random sampling of 70% of the data between 1985 to 2019. Then we assessed the performance of the models on a test set and on a validation set. The test set is made of the 30% remaining data available between 1985 and 2019 whereas the validation set are occurrences and covariates observed over 2020 and 2021 as a whole. None of the models are trained with data from 2020 and 2021, which makes this exercise a true out-of-sample experiment.

**Metrics**

To assess the ability of the models to predict flood occurrence based on new data, we calculated two performance metrics, that is, the area under the receiver operating characteristic (ROC) curve and the area under the precision-recall (PR) curve. Those two metrics are calculated globally (Canada and United States) and over PL2 watersheds to determine the regional performance of the predictions. The global metrics are shown below whereas the regional metrics are provided as part of the Supplementary Information (SI). The SI is an interactive HTML application generated by a R Markdown.

The ROC curve plots the true positive rate (also called sensitivity or recall) as a function of the false positive rate (one minus specificity). The closer the ROC curve is to the top-left, the better the sensitivity and the specificity of the model, and hence, the better its predictive capability. The area under the ROC curve is an aggregate measure that considers all values of false positive rates between 0 and 1. A true null classifier yields a straight diagonal line whose area below the ROC curve is 0.5.

Therefore, models with predictive capability will have areas above 0.5 whereas skilled models should yield an area above 0.7 (Buizza et al., 1999).

The PR curve is similar to the ROC curve, but is built on the precision and recall metrics instead. Precision is defined as the ratio of true positives over the sum of all predicted positive values. A high precision means that a model can correctly predict positive values without creating too many false positives. Recall is defined as the ratio of true positives over the sum of true positives and false negatives. A high recall means that a model predicts a small amount of false negatives. Therefore, a high value of the area under the PR curve means that a model can predict flood occurrence without misclassifying a significant

number of non-occurrences.

**Results**

Table 1 shows the performance of the 10 models using the area under the ROC and PR curves, under the test and validation sets.

| Models | Test | | Validation | |
|---|---|---|---|---|
| | ROC | PR | ROC | PR |
| GLM | 0.93494 | 0.70438 | 0.91441 | 0.59185 |
| GLM INTER | 0.93704 | 0.71066 | 0.91038 | 0.61194 |
| GLM 0US | 0.84867 | 0.35501 | 0.77166 | 0.19060 |
| GAM | 0.93736 | 0.71093 | 0.91454 | 0.61694 |
| GAM INTER | 0.93789 | 0.71175 | 0.91289 | 0.61625 |
| GAM 0US | 0.85244 | 0.36734 | 0.77331 | 0.20861 |
| RF | 0.96861 | 0.82717 | 0.89875 | 0.60338 |
| RF 0US | 0.95713 | 0.77644 | 0.75196 | 0.18475 |
| GBM | 0.94450 | 0.73724 | 0.91441 | 0.61791 |
| GBM 0US | 0.87634 | 0.44921 | 0.77179 | 0.20661 |

**Table 1.** Area under ROC and PR curves for each of the 10 models used. The test dataset consists of 30% of the observations from 1985 to 2019 (randomly sampled). The validation dataset consists of occurrences and covariates observed from 2020 and 2021. 0US: Models trained without upstream variables. INTER: Models trained with interactions between atmospheric variables.

We first observe that models that include upstream variables (covariates and flood indicator) show better predictive skill

than those that exclude these variables. This means that flood occurrence may exhibit spatial dependence in the form of spatial autoregression. In other words, we find that flood occurrence downstream is affected by flood occurrence and atmospheric variables both observed upstream. One may think however this could be due to the construction of the DFO flood polygons, but the mere fact that upstream variables are statistically significant shows that at least the shape and orientation of DFO flood polygons replicate the water flow between watersheds and this is important in the model. If polygons were randomly shaped

or too grossly drawn, it would not have been possible to detect any significant signal from upstream variables and inclusion of

upstream variables would yield weaker predictions. However regardless of the inclusion of upstream variables, all 10 models have predictive skill with an area under the ROC curve well above 0.85 in the test set and above 0.75 in the validation set, which is very good.

We also investigate the contribution to the predictive skill of the added interactions in the GLM and GAM models. We observe only a marginal gain in the test set (for both metrics) with a mixed signal in the validation set, that is, we remark a slight gain under the PR metric and small loss under the ROC metric. Given the added complexity tied to the extra 32 covariates and on the basis of parsimony, we decide not to further analyze these models.

For a fixed set of covariates, we compare the predictive skill of various classes of models; that is, we compare the GLM and GAM. Under the test set, the area under the ROC curve is similar for all four classes of models with upstream variables (0.93-0.97) but performance is better for the RF without upstream variables (0.96 vs 0.85-0.88). When we consider the area under the PR curve, the RF also outperforms other models, with (0.83 vs 0.70-0.74) or without upstream variables (0.78 vs 0.36-0.45). However, under the validation set, the models have much similar performance. For example, the area under the ROC (PR) curve is in the range of 0.90-0.91 (0.59-0.62) for all four models including upstream covariates, or 0.75-0.77 (0.19-0.21) without upstream variables.

The SI shows that the models perform very well across different regions of Canada and the United States but appear slightly weaker (but still skillful) in the South-Western United States (dry, desertic climates) and on the East Coast as well.

### 3.4 Key Determinants

In this section we analyze the key determinants of flood occurrence from six occurrence models, that is the GAM, RF and GBM, each with or without upstream variables. Those six models were deemed most relevant to further investigate on the basis of their predictive skill. Models with interactions have been excluded for parsimony and due to their lack of significant additional predictive ability, and the baseline GLM as well since it is encompassed in the GAM models. To explore the determinants, we trained the models again, but using 100% of the data available from 1985 to 2021.

Table 2 shows the top 10 most significant covariates for all six models according to the absolute t-value (GAM models), variable importance (RF models) and variable influence (GBM models). The sign of the coefficient is also provided for GAM models. The SI provides the full list of covariates, value of their coefficients and statistical significance (for both GAM models) and the variable importance/influence for the four RF and GBM models.

Without surprise, precipitation typically appears in the top 2 or 3 variables, with the 7-day precipitation being the most important across models. This is in line with Davenport et al. (2021) who found a strong link between 5-day precipitation and flood losses in the US. Under the GAM models that we can easily interpret, we observe that coefficients for precipitation covariates are all positive, meaning that flood occurrence probabilities increase with precipitation. Temperature covariates also appear significant but the sign of their coefficients is more difficult to interpret as we might capture seasonality and different hydrological regimes.

| GAM | GAM 0US | RF |
|---|---|---|
| IND_FLOOD_UPSTREAM + | Precip_Mean_7 + | IND_FLOOD_UPSTREAM |
| Precip_Mean_7 + | Precip_Mean_8_30 + | Precip_Mean_7 |
| Precip_Mean_8_30 + | Precip_Mean_61_12 + | Precip_Mean_61_120 |
| US_Precip_Mean_7 + | GDPPC + | Precip_Mean_31_60 |
| GDPPC + | GPW + | cly_pc_uav |
| GPW + | Temp_Min_Mean_61_120 + | Precip_Mean_8_30 |
| CLIMATE_KG +/- | Temp_Max_Mean_31_60 - | GPW |
| Temp_Max_Mean_31_60 - | snd_pc_sav - | Temp_Max_Mean_61_120 |
| US_Temp_Min_Mean_61_120 - | Precip_Mean_31_60 + | Temp_Min_Mean_61_120 |
| sgr_dk_sav - | CLIMATE_KG +/- | Temp_Max_Mean_31_60 |

| RF 0US | GBM | GBM 0US |
|---|---|---|
| Precip_Mean_7 | IND_FLOOD_UPSTREAM | CLIMATE_KG |
| Precip_Mean_61_120 | CLIMATE_KG | Precip_Mean_7 |
| cly_pc_sav | US_Precip_Mean_7 | snd_pc_sav |
| Precip_Mean_31_60 | Precip_Mean_7 | GDPPC |
| Precip_Mean_8_30 | GPW | cly_pc_sav |
| Temp_Max_Mean_61_120 | cly_pc_uav | GPW |
| Temp_Min_Mean_61_120 | US_Precip_Mean_61_120 | Precip_Mean_61_120 |
| Temp_Max_Mean_31_60 | snd_pc_uav | Precip_Mean_31_60 |
| Temp_Min_Mean_7 | GDPPC | Temp_Max_Mean_31_60 |
| Temp_Max_Mean_8_30 | water_ext | Precip_Mean_8_30 |

**Table 2.** Top 10 most significant covariates for six occurrence models. CLIMATE_KG is a factorial variable, thus we encompass each variable of the climate together.

For all models with upstream variables, the most significant one is the upstream flood indicator which points to how flood occurrence may be spatially correlated to upstream basins. It is also worth mentioning that in the GAM and GBM models, upstream variables such as precipitation and temperature also appear in the top 10.

Socioeconomic variables also appear significant across all models. Population per capita (GPW) is within the top 10 for 5 out of 6 models and GDP per capita (GDPPC) for 4 out of 6 models. In the GAM models, both of these variables have a positive influence on flood occurrence. Although wealth and vulnerability should be inversely proportional, it appears that population density and wealth better capture the overall impact of urbanization on flood occurrence than the urban extent land use variable.

Finally, the GBM models differ from other models by putting a greater emphasis on soil type and the Koppen-Geiger climate classification. Because these variables are static over time, it appears the GBM focused on leveraging the basin characteristics rather than capturing the dynamic aspect of atmospheric variables. The GBM models should therefore be less responsive to their daily dynamics, but better at distinguishing flood occurrence from one basin to the other.

## 4 Impact Model

In this section, we calculate the impact of a flood, describe the statistical and machine learning methods with their covariates, analyze the performance of the various impact models and summarize the key determinants of flood impact from our fitted models. The structure of this section is very much similar to that of Section 3.

### 4.1 Definition

In this paper, flood impact is measured in terms of population displaced and is based upon the DFO. According to Brakenridge (2022), population displaced "is sometimes the total number of people left homeless after the incident, and sometimes it is the number evacuated during the flood." In both cases, there is an economic loss: a house was either flooded (and therefore damaged), or water was close enough to a house such that there was a significant risk to human lives or disruption to human activities. As such, the number reported in DFO can be higher than the actual number of flooded people, which was also noted by Vestby et al. (2024). Overall, we find that population displaced is therefore a key driver of the socioeconomic impacts of a flood.

Whereas the DFO dataset directly provides the population displaced per event, we have to calculate how many people were potentially affected per event and per PL8 watershed since this is our basic geographical unit. We first started by approximating the number of people at risk of flooding per PL8 watershed. This is done by coupling a global map of the 100-year floodplain with a population dataset and aggregating the results per PL8 watershed. Flood maps are rarely available for free for such a large area, and hence we used the Flood Hazard Map of the World 0-100 year return period of the Joint Research Centre of the European Commission (Dottori et al., 2016). The resolution of the map is 1 km per 1 km which is helpful considering the small extent of some watersheds. We also used the Gridded Population of the World v4 dataset as described in Section 2.6. Instead of

redistributing displacements based upon the intersection of flood hazard and exposure, it would also be possible to redistribute displacements based upon gridded social vulnerability data such as age, income, etc. but this is left for future research.

Then, for each flood event in the DFO dataset, we took the spatial intersection between the PL8 boundaries and the event polygon, just like in the occurrence determination (Section 3.1), but with the goal of determining the number of people displaced per event and PL8 watershed. We assigned population displaced across intersected basins proportionally to the number of people at risk of flooding based upon the Flood Hazard Map of the World.

Finally, for the statistical and machine learning methods, we normalize the population displaced by the overall population of the watershed, and take the log (base $e$) of that ratio. The 504 events in the DFO therefore translate into 42,530 observations, that is, basin-events with at least 1 person at risk of flooding. The distribution of the observed log-ratio is provided in Figure 3. The average log-ratio is -8.54 and the standard deviation is 2.43. One has to be careful in the interpretation of descriptive statistics of log-ratios since the exponential of the average log-ratio is very different from the average of the ratio, the latter being the relevant statistic. Indeed, the average ratio of population displaced is 4.24% with a standard deviation of 46.4%. We also have a few observations above 0 which should not occur in theory. This happens in very small basins where the population displaced reported by the DFO is inconsistent with population data.

## 4.2   Methodology

Modelling flood impact is a typical regression problem where both statistical and machine learning approaches can be employed. We use the multiple linear regression, with possibly non-linear transformations of the covariates. Therefore, we apply the GLM and GAM for responses in $\mathbb{R}$ with a link function $g(x) = x$ in Equation 2. RF and GBM can also be used with $\mathbb{R}$-responses, and therefore, we have 10 models as well, based upon the same families of models and the same set of covariates as in Section 3.2.

## 4.3   Performance

As in Section 3.3, models were first fitted on a training set (random 70% of data available in 1985-2019) and then predictions were analysed on a test set (remaining 30% of the 1985-2019 data) and on a validation set (2020-2021, out-of-sample).

### Metrics

To assess the quality of the predictions, we computed two different metrics, that is the $R^2$ (larger is better, lower than 1) and the root mean square error (RMSE, lower is better). The $R^2$ or coefficient of determination is interpreted as the proportion of variance that is explained by the model (see James et al. (2021) Eq. 3.17) whereas the mean square error is the average of squared prediction errors (observed value minus its prediction). These two metrics are common with regression models. We show the global metrics (over Canada and US) below whereas regional metrics over PL2 watersheds are shown in the SI.

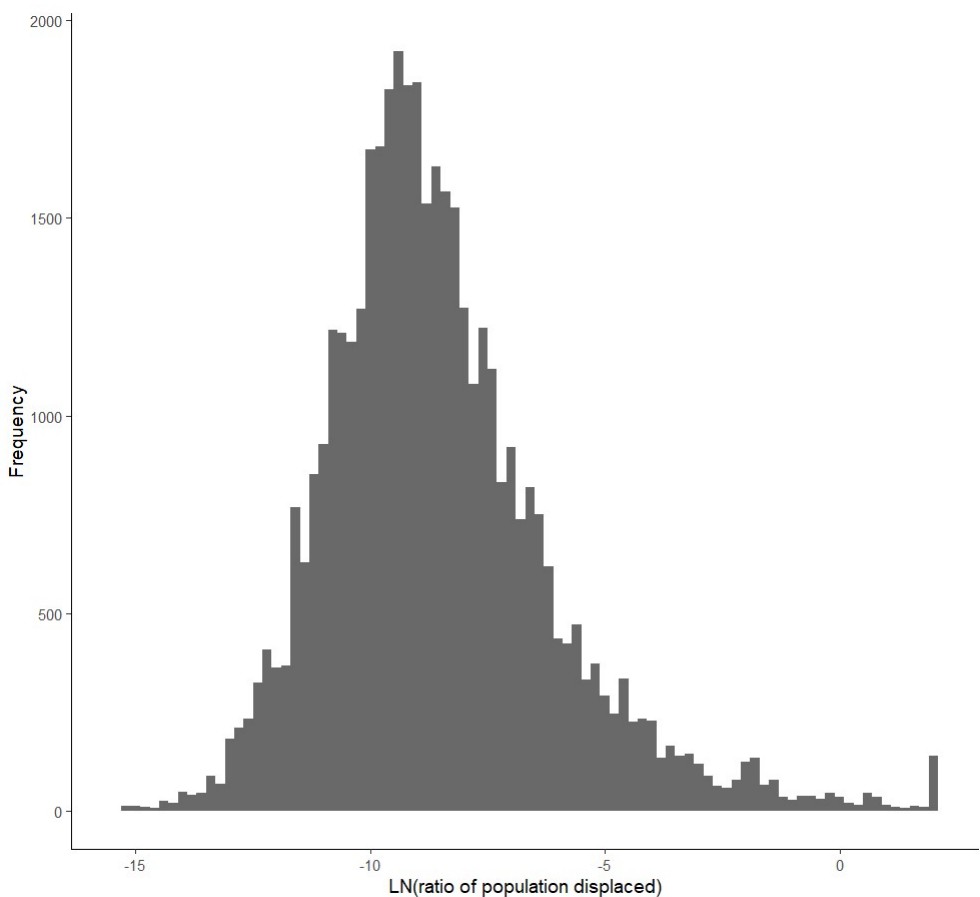

**Figure 3.** Histogram of the log-ratio of population displaced for each recorded flood occurrence

**Results**

Table 3 shows the performance of the 10 models using the $R^2$ and RMSE metrics, under both the test and validation sets.

First of all, we observe in Table 3 that interactions and upstream variables do not make much of a difference in predicting the impact of flooding. The GLMs and GAMs yield the largest RMSE and smallest $R^2$, although R-squareds in the range of 30% are decent given the complexity and the randomness of the problem at hand.

     In the test set, the RF and GBM yield very similar coefficients of determination and RMSE whereas the GAM and GLM provide a weaker performance. In the validation set however, the performance of the models is much more similar, with a slight

advantage to RF and GBM models ($R^2$ in 0.38-0.42) compared to GLM and GAM models ($R^2$ in 0.28-0.36). Models without upstream variables generally better perform in the validation set when compared to their equivalent with upstream variables. Although the performance of machine learning methods is slightly better in the validation set, we prefer the GAM without upstream variables for its interpretability and its very close performance to RF or GBM.

|  | Test | | Validation | |
| Models | $R^2$ | RMSE | $R^2$ | RMSE |
| --- | --- | --- | --- | --- |
| GLM | 0.27564 | 2.24257 | 0.32712 | 3.11864 |
| GLM INTER | 0.29542 | 2.21173 | 0.27601 | 3.23493 |
| GLM 0US | 0.26595 | 2.25751 | 0.35315 | 3.05773 |
| GAM | 0.27421 | 2.24477 | 0.33769 | 3.09405 |
| GAM INTER | 0.30325 | 2.19940 | 0.27434 | 3.23864 |
| GAM 0US | 0.26483 | 2.25924 | 0.35853 | 3.04498 |
| RF | 0.51128 | 1.84203 | 0.38154 | 2.98988 |
| RF 0US | 0.49730 | 1.86819 | 0.40169 | 2.94077 |
| GBM | 0.50806 | 1.84808 | 0.39017 | 2.96894 |
| GBM 0US | 0.50601 | 1.85192 | 0.42145 | 2.89180 |

**Table 3.** $R^2$ and root-mean-square error (RMSE) for each of the 10 models used. The test dataset consists of 30% of the observations from 1985 to 2019 (randomly sampled). The validation dataset consists of responses and covariates observed from 2020 and 2021. 0US: Models trained without upstream variables. INTER: Models trained with interactions between atmospheric variables.

The SI confirms the impact models perform well over five of six regions. In the Prairies however, the $R^2$ measured over the test set is about half of the one obtained over the other five regions. Comparing the quality of impact predictions over the validation set is however difficult as some regions had too few floods and impacts to validate the models over 2020 and 2021.

### 4.4 Key Determinants

We now analyze the key determinants of flood impact with six models, that is the GAM, RF and GBM, with or without upstream variables. Interactions were again excluded from such analysis, but we kept models with upstream variables for comparison. We trained the models again but using 100% of the data available from 1985 to 2021.

We see in Table 4 that the basin area (SUB_AREA) is always in the top three of the most significant covariates, whereas the urban extent appears for five out of six models. For both predictors, the sign is negative which might sound counter intuitive at first. However, we are modelling the log-ratio of population displaced, meaning that for a fixed flood severity and population at risk, there would be less population displaced for larger basin areas.

On the other hand, we see that the coefficients for GDP per capita and urban extent are also both negative (in the GAM 0US), which might reflect a decreased vulnerability over time. As such, we believe that socioeconomic variables proxied urbanization in the occurrence model whereas basin area, wealth and urban extent might have captured vulnerability in the impact model. This would lead for example to an increase of flood occurrence with greater urbanization, but a lower relative number of people displaced per flood, all else being equal. The DFO dataset also shows that population displaced per event has decreased over the last 40 years.

| GAM | GAM 0US | RF |
| --- | --- | --- |
| SUB_AREA - | SUB_AREA - | urban_ext |
| Temp_Max_Mean_7 + | Temp_Max_Mean_7 + | SUB_AREA |
| dor_pc_pva - | Temp_Min_Mean_7 - | crop_ext |
| Temp_Min_Mean_7 - | Precip_Mean_61_120 + | sgr_dk_sav |
| cly_pc_uav + | Precip_Mean_7 + | plain_ext |
| snd_pc_uav + | CLIMATE_KG +/- | forest_ext |
| US_Temp_Min_Mean_7 + | GDPPC - | Precip_Mean_7 |
| CLIMATE_KG +/- | dor_pc_pva - | GDPPC |
| US_Temp_Max_Mean_7 - | Temp_Min_Mean_8_30 - | Precip_Mean_31_60 |
| IND_FLOOD_UPSTREAM + | urban_ext - | Temp_Min_Mean_61_120 |

| RF 0US | GBM | GBM 0US |
| --- | --- | --- |
| urban_ext | urban_ext | urban_ext |
| SUB_AREA | CLIMATE_KG | CLIMATE_KG |
| crop_ext | SUB_AREA | SUB_AREA |
| sgr_dk_sav | crop_ext | crop_ext |
| Precip_Mean_7 | GDPPC | GDPPC |
| plain_ext | sgr_dk_sav | sgr_dk_sav |
| Precip_Mean_31_60 | plain_ext | Temp_Min_Mean_61_120 |
| forest_ext | Precip_Mean_7 | forest_ext |
| GDPPC | Temp_Min_Mean_7 | Temp_Min_Mean_7 |
| Temp_Min_Mean_61_120 | Temp_Min_Mean_61_120 | plain_ext |

**Table 4.** Top 10 most significant covariates for six impact models. CLIMATE_KG is a factorial variable, thus we encompass each variable of the climate together.

We also observe that covariates selected in the RF and GBM models tend to focus more on land use variables than on atmospheric variables. We therefore expect GAM models to be more responsive to changes in temperature and especially precipitation patterns than the former approaches. We remark that both precipitation variables are positive in the GAM 0US model, which therefore captures flood severity. All else being equal, more rain means more people are displaced. This analysis also confirms the usefulness and ease of interpretation of the GAM 0US model, therefore further supporting its use overall.

As with the occurrence model, the SI provides the full list of covariates, value of their coefficients and statistical significance (for both GAM models) and the variable importance/influence for the four RF and GBM models.

## 5    Applications

The flood model has static and dynamic inputs and therefore requires values for e.g., atmospheric variables to recover flood probabilities and the impact distributions. In this section we illustrate two potential applications. We first compute the modelled average annual population displaced in Canada and the US over 1980 to 2021 in each PL8 basin. We then analyze what-if scenarios where we shock precipitation to determine their overall impact on flood risk. Section 6 further discusses potential applications of the model within and outside the scope of this paper.

### 5.1    Population displaced due to flooding

We aim to compute the average annual number of people displaced by flooding in each PL8 watershed in the continental United States and Canada. Using observed covariates from 1985 to 2021 and the GAM model without upstream variables for both occurrence and impact components (see Sections 3 and 4), we have computed the predicted flood probabilities and population displaced for each hydrological year and PL8 basin. We then took historical averages to yield a single view of the average population displaced per PL8 basin.

Figure 4 shows the average annual number of people displaced for each PL8 watershed over the conterminous United States and Canada below $55°N$. Although the model runs over Alaska and the Canadian territories, those areas are not shown for conciseness. It is important to emphasize what is shown is the unconditional annual average, meaning that it includes the likelihood of flooding in the computations. In other words, it accounts for the fact there might be a 90% probability of zero people displaced due to flooding in a given basin. A higher value therefore means there is either significant flood hazard and/or a large number of exposed people. With approximately 20,000 basins displayed over the map, the model shows there are 0-120 people displaced every year on average per PL8 basin over 1985-2021.

Over the entire United States, the model annual average yields approximately 350,000 people displaced due to fluvial flooding over the Lower 48 in the period 1985-2021. There is in general greater flood risk in the Eastern US, especially around heavily urbanized areas (examples include New York, Philadelphia, Baltimore, Charlotte, Atlanta, New Orleans, Miami, Houston, St. Louis, Memphis, Nashville, etc.). The West Coast is also prone to flooding, especially in the states of OR (Portland), WA (Seattle) and CA. Overall, the spatial distribution of flood risk shown in Figure 4 is similar with the results obtained by the

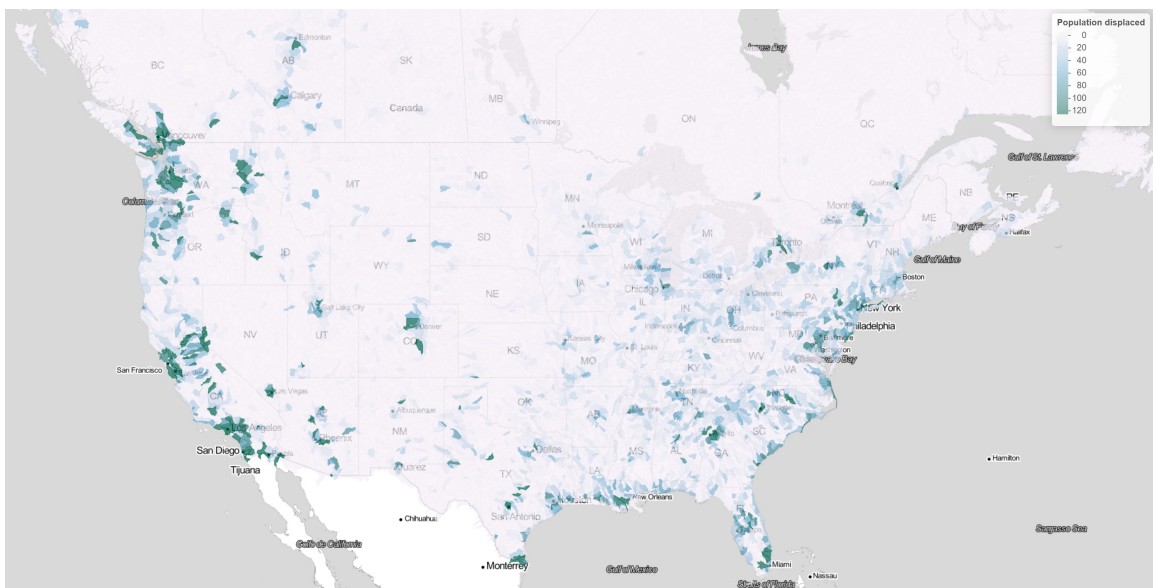

**Figure 4.** Average number of people displaced for each PL8 watershed over the conterminous United States and Canada between 1985 and 2021 using the GAM without upstream variables for both occurrence and impact models. Credits: Map produced using Leaflet with Stadia.StamenTonerLite layer.

First Street Foundation with Fathom (First Street Foundation (2020), page 10) who show the number of properties at substantial risk of flooding by county as of 2020.

In Canada, the modelled average annual number of people displaced over 1985-2021 is about 16,000. The map shows areas of greater flood risk along the St. Lawrence River (the Detroit/Windsor-Quebec City corridor), especially in Greater Toronto and Greater Montreal. Western Canada, especially Vancouver, Victoria, Calgary and Edmonton are also at significant risk of fluvial flooding as shown by Figure 4.

## 5.2 What-if scenarios

Another potential application of the model lies into the study of what-if scenarios, that is the analysis of the impact of a shock on any variable of the model (e.g., socioeconomic or atmospheric) to determine the overall effect on flood probabilities and/or population displaced. Examples include stress-testing insurance portfolios and the study of the economic impact of extra precipitation (due to climate change).

Table 5 shows the result of three scenarios on the flood occurrence probability, the impact of a flood (conditional distribution), and the overall annual average number of people displaced due to flooding (unconditional distribution). The areas of interest are Canada, United States and the sum of both countries. The shocks are +10%, +25% and +50% on the climatology of each precipitation covariate and uniformly applied over the areas of interest for illustration purposes.

A shock of +10% (+50%) is consistent with the expected increase of the maximum 5-day precipitation (standardized precipitation index) from the CMIP6 33-model average over a 2-degree warming scenario in many locations over Canada and the United States (IPCC WGI Interactive Atlas). We held all other dynamic variables fixed as of their respective years (temperature, and socioeconomic variables). For this experiment, we used the GAM model for both the occurrence and impact, without upstream variables.

| Area | Component/Shocks | Baseline | ×1.1 | ×1.25 | ×1.5 |
|---|---|---|---|---|---|
| | Occurrence probability | 0.1186 | 0.1290 | 0.1439 | 0.1669 |
| North America | Population displaced | 2 323 665 | 2 441 755 | 2 610 426 | 2 872 909 |
| | Combined | 275 683 | 315 063 | 375 634 | 479 575 |
| | Occurrence probability | 0.0336 | 0.0376 | 0.0437 | 0.0541 |
| Canada | Population displaced | 483 971 | 509 417 | 545 744 | 602 192 |
| | Combined | 16 269 | 19 182 | 23 889 | 32 559 |
| | Occurrence probability | 0.1833 | 0.1986 | 0.2202 | 0.2531 |
| USA | Population displaced | 1 880 063 | 1 974 858 | 2 110 274 | 2 321 088 |
| | Combined | 344 739 | 392 270 | 464 738 | 587 444 |

**Table 5.** Effect on flood risk of various shocks on precipitation. The mean of occurrence and of population displaced has been calculated over 1985 to 2021. The GAM model without upstream variables has been used for both flood occurrence and impact models.

The average annual modelled number of people displaced in Canada is 16,000 and 345,000 in the United States, averaged over 1985-2021. This corresponds to our baseline value, and is consistent with the values provided in the previous subsection. When we increase precipitation by 10%, the overall increase in the average annual number of people displaced increases by 18% in Canada, and 14% in the United States. In the extreme scenario where precipitation increases by 50%, the increase in the average annual number of people displaced is 100% in Canada and 70% in the US. It is therefore very interesting to observe that the model yields non-linear increases in the overall population displaced for each shock on precipitation.

## 6 Discussion and Conclusion

We have introduced a fluvial flood model for socioeconomic purposes that covers the entirety of Canada and the (continental) United States. The model is based upon flood occurrence and impact in terms of population displaced over Pfafstetter Level 8 watersheds. The spatial resolution is sufficiently high (above 31,000 basins over Canada and United States) for large scale socioeconomic studies while being fast to fit, adjust, predict and simulate from. Calibrated with the DFO, the occurrence and impact models aim to replicate flood events that are significant from a socioeconomic standpoint, rather than focusing on typical hydrological variables such as discharge or runoff.

We used various statistical and machine learning methods to fit the occurrence and impact components. We found that flood occurrence does exhibit spatial dependence upon upstream basins while this is not the case for flood impact. The interactions

introduced in the GLM and GAM models have virtually no impact on the quality of the predictions. We found that precipitation variables, especially 7-day precipitation, are the most significant covariates, in addition to land use and socioeconomic variables to predict flood occurrence. We found that flood impact, conditional upon a flood being observed, is driven less by atmospheric variables than socioeconomic variables, and basin and land use characteristics. While population density and GDP per capita both appear to increase flood occurrence as possible proxies for urbanization, GDP contributes to decrease flood impact as a likely proxy for vulnerability.

In general, the GAM model is the preferred approach for both the occurrence (binary response) and impact (continuous response) as the predictive capability is comparable to machine learning methods in the validation set (which is a purely out-of-sample analysis) and is easily interpretable. But as the performance of the three main classes of models is very often equivalent, one can also use model averaging taking the mean prediction from the three classes of models, which would in effect yield an ensemble model.

The flood model has static and dynamic inputs and therefore requires values for atmospheric variables to obtain flood probabilities and the impact distribution. We distinguish two types of applications related to predictions and simulations.

Predictions relate to applications where a limited number of scenarios are used as inputs to the fitted occurrence and impact models. It involves using, for example, ensemble means from long-range and seasonal weather forecasts as inputs to compute predicted flood probabilities and population displaced in each watershed of a given region or country. From a methodological standpoint, such predictions require similar computations than in Section 5.1. What-if scenarios and sensitivity analyses are also useful to understand the impact of a % increase in population displaced for scenarios where the average precipitation is shocked by +10%, +25% or +50% above its climatology. Such scenarios have been applied in Section 5.2. We found for example that an increase of 10% (50%) in precipitation yields an increase of population displaced of 18% (100%) in Canada and 14% (70%) in the US.

Because the model is fast to run with thousands of climate simulations such as outputs from a stochastic precipitation simulator, it is therefore possible to translate those with the model to obtain the full distribution of flood occurrence probabilities and impact. Upstream variables could therefore be included to obtain a full representation of local flood risk, as well as the counts of basins that may flood in a given year over a larger region. Spatial dependence in flooding impedes the ability of an insurer to diversify flood risk and needs to be taken into consideration for reserving and capital determination.

Full integration of climate models (regional climate models or statistically downscaled general circulation models) is also feasible, under the present or future climate, if one is interested in analyzing the impacts of climate change on population displaced due to flooding. But large scale simulation of the socioeconomic impact of flooding under the present or future climate is out of scope for this paper and is left for future research.

There are also model components we could improve in the future, notably the quantification of flood impacts. Leveraging satellite imagery from the Global Flood Database, it would be possible to more accurately measure and predict displacements and therefore, socioeconomic impacts (see e.g., Tellman et al. (2021); Vestby et al. (2024)). And instead of redistributing displacements over watersheds based upon flood hazard as we did in Section 4.1, one could also redistribute displacements

based upon gridded social vulnerability data. Again, this would improve the accuracy of predicted displacements from the impact model.

## Appendix A: Implementation of Machine Learning Methods

This section details the implementation of machine learning methods, namely the RF and GBM, and how hyperparameters have been determined.

RF are fitted using the package `ranger` in R (Wright et al., 2020). This package is coded in C++, which significantly increases the speed at which we can fit models with millions of observations. Some hyperparameters are required before we can fit a model: the number of trees inside the RF (100, 500, 1000), the number of splits per tree (10,20,40,80,0), and the number of variables that we sample for each tree (6, 8, 10, 12, 14). To note, a value of 0 in the number of splits means that each tree is going to be fully developed.

To determine the best set of hyperparameters, we split the original training dataset such that 70% of the data is fitted with a specific set of hyperparameters. Among hyperparameters tested, we pick the set that maximizes the AUC of the ROC curves (for the occurrence model) and $R^2$ (for the impact model) in the 30% remaining data.

GBM are fitted using the well-known package `gbm` in R (Greenwell et al., 2019). Some hyperparameters are required before we can fit a model: the number of decision trees (100, 500, 1000, 2000), the complexity of each tree (1, 2), and the learning rate (0.001, 0.005, 0.01, 0.05, 0.1). The complexity of each tree is about the inclusion of interaction effects or not. If we do not want interaction effects, we are limited to only one split. Otherwise, we permit each tree to split twice which allows, at most, two variables to contribute to the tree. The same methodology as the random forests has been applied to determine which set of hyperparameters to choose.

*Data availability.* All datasets used in this study are openly available and can be downloaded directly from their source's websites. For convenience, the Supplementary Information is a R Markdown HTML file that includes direct links to the datasets, as well as the most complete outputs and validations of the occurrence and impact models. It is available on Zenodo with DOI 10.5281/zenodo.10201818.

*Author contributions.* Conceptualization : MG, MB, DAC;
Data curation : MG;
Formal analysis : MG, MB, JB;
Funding acquisition : MB, SR;
Investigation : MG, MB;
Methodology : MG, MB, DAC;
Project administration : MB, SR;
Resources : MG, MB;

Software : MG;

Supervision : MB;

Validation : MG, MB, JB;

Visualization : MG;

Writing – original draft preparation : MG, MB;

Writing – review & editing : MG, MB, DAC, JB, SR;

*Competing interests.* The authors declare they have no *direct* competing interest.

However, M. Boudreault received funding from Mitacs through the Accelerate program as a Professor at Université du Québec à Montréal with contributions from The Co-operators General Insurance Company and the Canadian Institute of Actuaries.

585 M. Grenier was a graduate student at Université du Québec à Montréal when most of the work was accomplished and is now employed by The Co-operators General Insurance Company.

D. A. Carozza was employed by The Co-operators General Insurance Company when the work was accomplished.

J. Boudreault and S. Raymond are currently employed by The Co-operators General Insurance Company.

*Acknowledgements.* Manuel Grenier and Mathieu Boudreault would like to thank The Co-operators and the Canadian Institute of Actuaries 590 for their financial support through the Mitacs Accelerate Program. Manuel Grenier would like to thank the Natural Sciences and Engineering Research Council of Canada (NSERC), the Fonds de recherche du Québec - Nature et technologie and the Canadian Institute Actuaries for their graduate scholarships. Mathieu Boudreault recognizes the financial support from the NSERC Discovery Grant Program.

The authors would like to thank Jean-Philippe Boucher and Philippe Lucas-Picher for their comments on earlier versions of this work.

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
