# Peer review of "Flood Occurrence and Impact Models for Socioeconomic Applications over Canada and the United States"

_EGUsphere, 2023_

## Author Comment (AC1)

**EGUSPHERE-2023-3039**
**Response to referees**

**Response to reviewer # 1**

We thank the referee for having reviewed the paper and for having provided comments. NHESS/Copernicus mentions that a revised manuscript should not be prepared at this stage. We include below our answers to each comment, as well as to how we would change the manuscript in the revision. The comments are replicated integrally and our answers are written in **blue and bold** font immediately underneath.

**\*\*\*\*\*\*\*\*\*\***

This manuscript contributes by modelling flood occurrences and their impacts using statistical and machine learning methods. The paper demonstrates high-quality research based on the methodology's effectiveness in modeling floods and predicting not only affected areas but also the displacement of populations. It presents important characteristics related to flood occurrence and impact that will inspire future research, especially when extrapolating the methodology to other regions. The paper is well-presented overall, containing crucial information that is carefully provided.

> **Thanks very much for the positive review.**

Additional comments:

1. Check for spacing between numbers and units throughout the paper to correct numerous typos in this regard.

   > **We indeed found instances of such typos at L158, L176 and elsewhere.**

   > **⇒ We will double check the manuscript to correct these typos. If the paper is accepted for publication, the typesetting stage shall correct any of these remaining typos as well.**

2. Provide a more detailed explanation of what is meant by "Modelling flood occurrence is akin to a classification problem." Be specific.

   > **We defined flood occurrence over L188-L190 as the intersection of a DFO polygon with a PL8 watershed. There are therefore two possible outcomes: flood (intersects) or no flood (does not intersect). This is different from hydrological approaches that typically define flooding using discharge or runoff and whose flood frequency is expressed in return periods.**

**In data science or machine learning, there is a classification problem when the variable we aim to explain is categorized in 2 or more classes. In our case, flood or no flood are 2 classes. As such, flood occurrence as defined in the manuscript is a classification problem. Moreover, usage of the word "akin" is not necessary in the quoted sentence.**

**⇒ We suggest explicitly stating that the variable we aim to explain has 2 classes (flood occurrence or no flood occurrence) which yields a classification problem. We would therefore modify the first sentence of Section 3.2 accordingly.**

3.  Represent "s" as a sub-index in "βs".

    **The "s" was meant to be the plural form of β as in betas.**

    **⇒ We suggest using formal mathematical notation to define the betas along the lines of (using LaTeX) : and $\beta_0, \beta_1, \beta_2, …$ are the coefficients of the model.**

4.  L497-L500. Reconsider the phrasing of the conclusion regarding what-if scenarios percentages. This is not very clear in the conclusions as it is in Section 5.2.

    **⇒ We suggest replacing the current sentence with (changes in italics):**
    **"What-if scenarios and sensitivity analyses are also useful to understand the impact of a % increase in population displaced for scenarios where *the average precipitation is shocked by +10%, +25% or +50% above its climatology*."**

---

## Author Comment (AC2)

**EGUSPHERE-2023-3039**
**Response to referees**

**Response to reviewer # 2**

We thank the referee for having reviewed the paper and for having provided comments. NHESS/Copernicus mentions that a revised manuscript should not be prepared at this stage. We include below our answers to each comment, as well as to how we would change the manuscript in the revision. The comments are replicated integrally and our answers are written in **blue and bold** font immediately underneath.

**\*\*\*\*\*\*\*\*\*\***

The manuscript presents a set of flood occurrence and impact models that may be applied to a large-scale spatial extents at low cost. The impact model focuses on population displacement, a key metric of concern for future climate change scenarios. While the structure and methodology is presented in a clear manner, improvements in framing and clarification of some concepts and methodological choices would enhance the manuscript.

> **Thanks very much for the positive review. Indeed, we believe that the clarifications below would improve our manuscript.**

1. Definition: It would be good to clarify what is captured by the outcome variable "population displacement". Does it include both temporary and permanent displacement? Does it include evacuations from areas that were not necessarily directly affected by floods? What are the implications of these different definitions for the manuscript's consideration of displacement as a proxy for socioeconomic impacts?

   > **The Dartmouth Flood Observatory (DFO) defines "Population displaced" as the number of people left homeless after the incident or the number of people evacuated during the flood (https://floodobservatory.colorado.edu/Archives/ArchiveNotes.html).**

   > **It is our understanding that this includes both temporary and permanent displacements, and also evacuations from areas that were not directly affected. The number reported in DFO can therefore be higher than the actual number of flooded people. This was also explicitly mentioned in Vestby et al. (2024): "Depending on whether predisaster evacuation and impacts of compounding and cascading hazards are accounted for, the number of displaced persons registered in the DFO catalogue may be many times higher than the number of residents in the flood-affected area."**

   > **In all these cases, it is clear there is an economic loss, and therefore DFO's number of displaced people is a reasonable proxy for socioeconomic impacts. Either a house was completely flooded (and therefore damaged), or water was close enough to a house that there was a significant risk to human lives. For example, damage to a flooded property**

**and losses suffered because of an evacuation (even if not flooded) are covered by typical flood insurance policies or through public disaster financial assistance programs.**

**⇒ We suggest adding the aforementioned definition of "Population displaced" in the first paragraph of Section 4.1 ([Impact Model] Definition) along with more justification as to how such definition relates to socioeconomic impacts. We suggest quoting Vestby et al. (2024) to contrast the number of flooded people vs displacements reported by DFO.**

2. DFO polygons: The motivation for using DFO polygons to identify flood occurrence is unclear. Others in the literature have utilized satellite-based flood extents that are much higher resolution. (e.g. Vestby et al 2024)

**At the time we conducted the study, few historical floods had high-resolution inundation extents, especially in the DFO. We quote Mester et al. (2023): "Until recently, only a limited number of satellite observations of past floods were publicly available, mainly provided by the archive of the Dartmouth Flood Observatory (DFO), and the United Nations Operational Satellite Applications Programme (UNOSAT). With the release of the Global Flood Database (GFD), a product based on the DFO, an unprecedented inventory of satellite imagery is now available, including 913 large flood events from 2000 to 2018." Note that the GFD was also used by Vestby et al. (2024).**

**Mester, B., Frieler, K. & Schewe, J. Human displacements, fatalities, and economic damages linked to remotely observed floods. Sci Data 10, 482 (2023). https://doi.org/10.1038/s41597-023-02376-9**

**With over 5,000 events listed globally in the DFO between 1985 and 2021, it is clear that even the GFD only covers a fraction of historical DFO events. This is further supported by Vestby et al. (2024): "The GFD data represent a subset of all DFO-recorded floods in this period, since adverse meteorological conditions and complex topography sometimes prevent remote sensing."**

**The DFO polygons, although more rough, provide a good indicator of where and when flooding occurred, and are all accompanied by displacement numbers for each event since 1985. Each event is backed by at least one source to confirm the flood event (news, authorities, satellites).**

**That being said, usage of DFO polygons vs satellite-based flood extents depends on use cases. For the occurrence model presented, we aim to determine the annual flood occurrence probability per PL8 watershed. Underreporting of flood events is therefore a concern. It is also important to keep in mind that a PL8 watershed is typically a large area. On the other hand, satellite imagery is useful to determine occurrence and impacts at much higher resolution, e.g., at the neighborhood-level or city-level. But high-resolution inundation extents may be lacking or may be incomplete when dense clouds were present during the flood (see quote from Vestby et al. (2024) as well).**

**Keeping in mind that the focus of the occurrence model is on PL8 watersheds, we preferred the DFO polygons for two reasons. First, a flood event is more likely to be reported in the DFO since it is also based upon news and authorities reports, even when satellite imagery is lacking. Moreover, having more than 35 years of data is also helpful to detect statistically significant patterns and trends in the occurrences, which is a key use case for both the occurrence and impact models.**

**We also recognize the added value of satellite-based flood extents to more accurately estimate human displacements and socioeconomic impacts at higher resolution when a flood has occurred. The recently released GFD is therefore a welcome addition.**

**⇒ We suggest adding a sentence or two in Section 2.1 ([Data] Observed Floods) to better justify why we used DFO along the lines of our response above. We also propose concluding the manuscript with a suggestion of using the GFD in addition to DFO in the future to more accurately measure displacements and socioeconomic impacts at high resolution while citing Vestby et al. (2024).**

3. Exposure vs Vulnerability: The authors assign population displaced per watershed based on the total population exposed, which does not address factors that make certain groups more vulnerable to displacement. How might considerations of differential vulnerability impact the models' predictions?

**If the reviewer is referring to Section 4.1, then yes, we did redistribute the observed population displaced over intersecting watersheds according to the 100-year return period floodplains (Flood Hazard Map of the World of the Joint Research Centre of the European Commission) and a gridded population dataset. The idea was to make sure to redistribute displacements in accordance with the watersheds most vulnerable to flooding, when vulnerable means being located in the 100-year floodplain.**

**If one were to use a more societal definition of vulnerability that includes, e.g., age and/or income for example, a similar methodology could be applied to redistribute displacements, as long as there exists a gridded dataset with various vulnerability levels.**

**As to how it would impact the models' predictions is certainly an interesting question. Redistributing displacements differently would affect the spatial distribution of displacements given there is a flood (but would not affect the occurrences), and would tend to replicate the historical social vulnerability of flooding.**

**⇒ We suggest adding a sentence or two about different redistribution schemes for displacements in Section 4.1 - 3rd paragraph along the lines of our response above. We also propose concluding the manuscript by describing this idea for future research if such data is available.**

4. Model variables: The motivation for adding 32 interactions across the climatic variables is unclear (260-263). Please describe in more detail the key interactions of interest that are being tested.

   **Adding interactions allows us to determine what happens to flood occurrence (and impact) when two variables go in the same direction or in opposite directions. For example, when there is precipitation and temperature is below 0°C, then snow accumulation clearly changes flood dynamics. Another example is when heavy rain follows a drought; that might increase flood probabilities as well.**

   **The occurrence and impact models are based upon three atmospheric variables: maximum and minimum temperatures and precipitation. For each atmospheric variable, we computed averages over four non-overlapping time frames: short term (1-7 day prior to flood occurrence), mid term (8-30 days), long term (31-60 days), and very long term (61-120 days). This is described in Section 2.4.**

   **Then, for each time frame, we looked at the potential interactions between the three atmospheric variables (one by one, and all together). This gives us 16 possible interactions. To these 16, we add those from the upstream watershed (same combinations but from upstream) which gives us another 16 possible interactions. Therefore, the model with interactions (INTER suffix) has 32 interactions in total.**

   **⇒ We suggest rewriting the corresponding paragraph (around line 260) to add more details about interactions along the lines of our response above.**

5. (301-304) The validation against DFO polygon boundaries is confusing, given the polygons were used to provide flood occurrence data in the first place. Please clarify?

   **The purpose of L301-L304 is not to validate the spatial distribution of occurrences (ones and zeroes) against the DFO polygons. That would indeed be redundant since the DFO polygons were used to generate occurrences in the first place.**

   **The objective of this analysis is rather to determine whether there is spatial autoregression in occurrences or in other words, whether flood occurrence downstream is affected by flood occurrence and atmospheric variables both observed upstream. We found this was indeed the case. One may think however this could be due to the construction of the DFO polygons, but the mere fact that upstream variables are statistically significant shows that at least the shape and orientation of DFO polygons replicate the water flow between watersheds and this is important in the model. If polygons were randomly shaped or too grossly drawn, it would not have been possible to detect any significant signal from upstream variables.**

   **⇒ We suggest adding more details within the paragraph around lines 301-304 to clarify the analysis and results along the lines of our response above.**

6. (171) "Given how floods are reported in the DFO database, we include socioeconomic variables such as population and wealth in the proposed flood model." What does this mean?

**For a flood event to be included in the DFO flood database, it must impact a population and/or cause economic losses. It is therefore natural that observed flood occurrences (derived from the DFO polygons) are also driven by socio-economic variables such as population and wealth (in addition to precipitation and temperature), therefore justifying their addition in the models.**

**⇒ We suggest modifying the first paragraph of Section 2.6 (Socioeconomic Variables) along the lines of our response above.**

Vestby, J., Schutte, S., Tollefsen, A. F., & Buhaug, H. (2024). Societal determinants of flood-induced displacement. Proceedings of the National Academy of Sciences, 121(3), e2206188120. https://doi.org/10.1073/pnas.2206188120

---

## Author Response (AR2)

**EGUSPHERE-2023-3039**
**Response to reviewer # 2**

The authors' clarifications have greatly improved the manuscript.

**We are happy that the reviewer found the corrections and clarifications to have greatly improved the manuscript.**

One further clarification suggestion is as follows: the authors note that socioeconomic variables (GDP per capita and population density) are included in the flood occurrence model because DFO flood entries are recorded only when significant damages or fatalities are reported. However, this seems to suggest the model predicts whether a flood event was recorded in the DFO, rather than all occurrences of flood events. It would be good to further clarify the rationale for including these variables in the context of a flood prediction model, and potential implications for future users of the model.

**The model does not predict whether an event was recorded in the DFO. We rather use the DFO to determine historical flood occurrences and to calibrate various flood occurrence models. As a result, the presented flood occurrence models would predict a flood occurrence only if environmental and socioeconomic conditions are both favorable to impactful flooding. An overflow of water in a random location that yields no economic loss, disruption nor displacement is unlikely to be recorded in the DFO and would have very limited socioeconomic impact. This was the intent of the model from the beginning and this is how the model should be used.**

**As shown by the latexdiff file provided, we added clarifications in three different locations of the manuscript, that is the Introduction, early in Section 3.2 and in the Discussion and Conclusion.**